# Modeling Method for Aerobic Zone of A$^2$O Based on KPCA-PSO-SCN

**Wenxia Lu, Xueyong Tian \*, Yongguang Ma, Yinyan Guan, Libo Liu and Liwei Shi**

School of Environmental and Chemical Engineering, Shenyang University of Technology,
Shenyang 110870, China; lu_wx116@smail.sut.edu.cn (W.L.); 13694131901@163.com (Y.M.);
guanyinyan@sut.edu.cn (Y.G.); 15524172565@163.com (L.L.); 15032955026@smail.sut.edu.cn (L.S.)
\* Correspondence: tianxueyong@sut.edu.cn; Tel.: +86-186-0243-8801

**Abstract:** Sewage treatment plants face significant problems as a result of the annual growth in urban sewage discharge. Substandard sewage discharge can also be caused by rising sewage treatment expenses and unpredictable procedures. The most widely used sewage treatment process in urban areas is the Anaerobic–Anoxic–Oxic (A$^2$O) sewage treatment process. Therefore, modeling the sewage treatment process and predicting the effluent quality are of great significance. A process modeling method based on Kernel Principal Component Analysis–Particle Swarm Optimization–Stochastic Configuration Network (KPCA-PSO-SCN) is proposed for the A$^2$O aerobic wastewater treatment process. Firstly, eight auxiliary variables were determined through mechanism analysis, including Chemical Oxygen Demand (COD) and ammonia nitrogen (NH$_4^+$) and nitrate nitrogen (NO$_3^-$) of influent water, pH, temperature (T), Mixed Liquor Suspended Solid (MLSS), Dissolved Oxygen (DO) and hydraulic residence time (HRT) in the aerobic zone. Dimensionality reduction was carried out using the kernel principal component analysis method based on the Gaussian function, and the eight-dimensional data were changed to five-dimensional data, which improved the running speed and efficiency of subsequent models. Then, according to the advantages of the particle swarm optimization algorithm, such as low calculation cost and fast convergence, combined with the advantages of stochastic configuration network general approximation performance, the PSO-SCN model was established to predict the three water quality indexes of effluent COD, NH$_4^+$, and NO$_3^-$ for the aerobic zone. The experimental results proved the effectiveness of the model. Compared with classic water quality prediction algorithm models such as SCN, PSO-BP, RBF, PSO-RBF, etc., the superiority of the PSO-SCN algorithm model was demonstrated.

**Keywords:** A$^2$O; kernel principal component analysis; particle swarm optimization; stochastic configuration network; process modeling

## 1. Introduction

With the acceleration of urbanization and the improvement of people's living standards, the amount of urban sewage has increased year by year in China. According to this survey [1], the annual discharge of sewage in China has increased from 41.67 billion cubic meters in 2012 to 62.5 billion cubic meters in 2021. In order to solve the problem of water pollution and water resource shortage, the treatment and reuse of urban sewage is particularly important. Urban sewage treatment removes pollutants in sewage through biological treatment technologies, such as adsorption, decomposition and oxidation of microorganisms, which is an important means to maintain sustainable utilization of water resources [2,3]. Common biological treatment technologies include the activated sludge process and biofilm process [4]. The activated sludge process has been widely used in urban sewage treatment because of its high efficiency and economic advantages, the most representative of which is the A$^2$O process with the activated sludge process as the core [5].

The main function of the A$^2$O sewage treatment process is denitrification and phosphorus removal. Its core is the activated sludge method, which degrades organic matter in

sewage through microorganisms' adsorption, decomposition, and oxidation. At the same time, nitrogen, phosphorus, and other elements in sewage are removed through a series of biochemical reactions to achieve relevant discharge standards. In the whole process of $A^2O$ wastewater treatment, the biochemical effect of the aerobic zone plays a crucial role. The biochemical reactions at this stage are the most complex, and its main functions are the nitrification of ammonia nitrogen, the absorption of phosphorus, the degradation of organic matter, etc. The sewage treatment capacity in the aerobic zone has an important impact on the effluent quality. Therefore, modeling the aerobic zone can not only timely reflect the sewage treatment effect of its related biochemical zone, but also predict the effluent quality of the $A^2O$ process, to play a warning role in whether the effluent quality meets the standard, which is of great significance. At the same time, the aeration energy consumption of the aerobic zone accounts for 70% of the total energy consumption of the entire $A^2O$ process. The modeling of the aerobic zone also provides a basis for further optimization of aeration volume and saving of aeration energy consumption in the $A^2O$ process.

The reaction of $A^2O$ process is complicated and there are many influencing variables, so it is difficult to accurately predict the effluent quality. At the same time, the important parameters of effluent water quality (such as COD, ammonia nitrogen, etc.) can only be obtained after the completion of the process, which has a serious lag, which will lead to substandard sewage discharge.

The traditional modeling methods of the sewage treatment process are mainly based on the mechanism method, for example, the activated sludge kinetics equation based on mathematical expression proposed by Eckenfelder [6] and Mckinney [7] et al. The ASMs series models [8–12] and BSMs series models [13–15], as well as simplified and modified models proposed by the International Water Pollution Research and Control Association, which have been influential and are still in use today, all use the mechanism approach for process modeling. In the face of the complicated flow system of sewage treatment, it is difficult to describe the whole sewage treatment process in detail with the mechanism model, so it is imperative to establish and use a more accurate prediction model. With the development of artificial intelligence technology, the mechanism model is gradually replaced by the artificial neural network model because of its shortcomings such as uncertainty, strong nonlinear, and large time delay. Neural networks can learn the data, excavate the rules between the data, adjust the accuracy of the model by using the error mechanism, etc., and obtain the optimal function and optimal solution, which has strong expression and fitting ability. Therefore, using neural network methods to model sewage treatment processes has become a research hotspot.

For the process modeling of sewage treatment, Wan et al. [16] established a soft sensor model for Suspended Solids (SS) and COD in effluent using an adaptive fuzzy inference system based on neural networks. Han et al. [17] used the self-organizing radial basis function (SORBF) neural network method to predict the changes in the Sludge Volume Index (SVI). The experimental results show the effectiveness of this method. Qiao et al. [18] proposed a Repair Radial Basis Function (RRBF) to predict effluent COD, which has been proven to be effective in experiments. Yang et al. [19] combined PSO with a Nonlinear Auto-Regressive model with Exogenous Inputs (NARX) to establish a dynamic PSO-NARX model for predicting the COD and total nitrogen (TN) in effluent. The experimental results showed that the model had high prediction accuracy. Kusiak, Caneta, Bagheri et al. [20–22] used Multi-Layer Perceptron (MLP) to establish models for energy consumption and sewage flow rate, COD, SVI, etc., and achieved good prediction results in the experiment.

However, the neural network modeling methods used in the above studies have problems such as high manual intervention, limited universal approximation ability, and susceptibility to falling into local optima. To address the above issues, this article proposes a process modeling method for the aerobic zone of the $A^2O$ process based on KPCA-PSO-SCN. This method first uses KPCA to reduce the dimensionality of the data, solving the drawbacks of high computational complexity and slow computational speed caused by

high dimensionality and then uses PSO to optimize SCN. The effluent COD, $NH_4^+$, and $NO_3^-$ in the aerobic zone were predicted.

## 2. Selection and Treatment of Auxiliary Variables in A$^2$O Aerobic Zone

The main functions of the aerobic zone of the A$^2$O wastewater treatment process are nitrification of ammonia and nitrogen, absorption of phosphorus, degradation of organic matter, etc. In the actual process, the treatment effect of ammonia nitrogen has a huge impact on the effluent. Therefore, this article focuses on the nitrification of ammonia nitrogen and organic matter degradation in the A$^2$O aerobic zone for research and modeling. The sewage treatment process in the A$^2$O aerobic zone is shown in Figure 1.

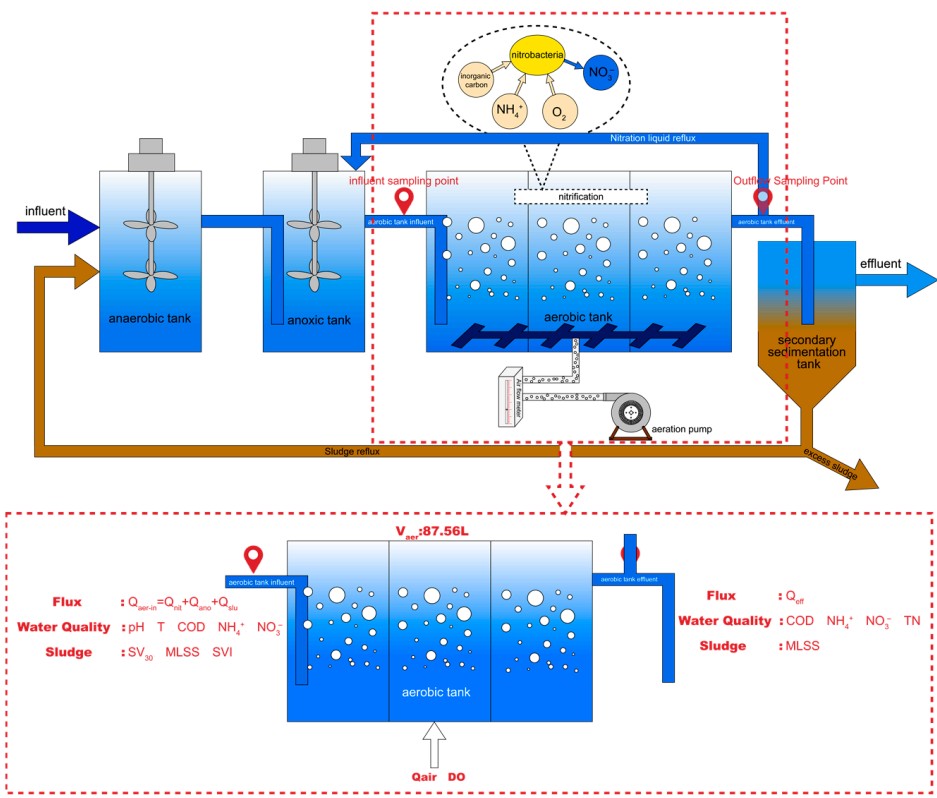

**Figure 1.** A$^2$O aerobic zone sewage treatment process diagram.

- Nitration of ammonia nitrogen;

The sewage containing ammonia nitrogen enters the aerobic tank after passing through the anaerobic tank and anoxic tank, and then oxidizes ammonia nitrogen and produces a large amount of nitrate nitrogen to form a nitrifying liquid under the action of nitrifying bacteria, an inorganic carbon source and oxygen. After external reflux, the nitrated liquid enters the anoxic tank for the next reaction.

- Degradation of organic matter.

In the aerobic tank, microorganisms use organic matter as a carbon source and energy. Under the action of oxygen, the organic matter is decomposed into carbon dioxide and water, promoting the growth of microorganisms in activated sludge.

According to the research contents and objectives, glucose and milk powder were selected as carbon sources and ammonium sulfate $((NH_4)_2SO_4)$ as nitrogen sources to simulate urban sewage as much as possible. When the ratio of carbon to nitrogen in sewage is 100:5, the COD in sewage is 500 mg/L and the nitrogen content is about 25 mg/L. The specific ratio is shown in Table 1.

**Table 1.** Sewage preparation ratio.

| | Carbon Source | | Nitrogen Source | |
|---|---|---|---|---|
| | Raw Material (g/L) | COD Content (mg/L) | Raw Material (g/L) | Nitrogen Content (mg/L) |
| Glucose | 0.3 | 250 | - | |
| Milk powder | 0.3 | 250 | Trace neglect | |
| $(NH_4)_2SO_4$ | - | | 1.18 | 25 |
| Total | - | 500 | - | 25 |

*2.1. Data Collection and Selection*

There are many factors that affect the effectiveness of sewage treatment. In order to establish an accurate and reliable process model, starting from the mechanism reaction of the sewage treatment process, the factors that have the greatest impact on output variables are selected [23,24]. Based on the analysis of microbial growth, water quality requirements of sewage treatment and factors affecting water quality in the mechanism reaction process, this paper collected data on water quality indexes of influent and effluent water in aerobic zones, sludge indexes, and water quality indexes in the pond, respectively (collection points are shown in Figure 1), and finally determined eight variables as the input of the model and three variables as the output of the model. The acquisition variables are described in Table 2.

**Table 2.** Collection variable description.

| | | Indicator Symbols | Unit | Description |
|---|---|---|---|---|
| Model Input Variables | Aerobic tank influent water quality indicators | $I_{COD}$ <br> $I_{NH4}^+$ <br> $I_{NO3}^-$ | mg/L <br> mg/L <br> mg/L | COD concentration of influent water <br> Influent ammonia nitrogen concentration <br> Concentration of nitrate nitrogen in influent water |
| | Water quality indicators in aerobic tanks | pH <br> T <br> MLSS <br> DO | - <br> °C <br> g/L <br> mg/L | - <br> Temperature <br> Mixed liquor suspended solid <br> Dissolved oxygen concentration |
| | other | HRT | h | hydraulic retention time |
| Model Output Variables | Aerobic tank effluent water quality indicators | $E_{COD}$ <br> $E_{NH4}^+$ <br> $E_{NO3}^-$ | mg/L <br> mg/L <br> mg/L | COD concentration of effluent <br> Ammonia nitrogen concentration in effluent <br> Nitrate nitrogen concentration in effluent |

Selection of eight auxiliary variables:

1.  COD;

COD is the measurement of the amount of reducing substances that need to be oxidized in a water sample using chemical methods. To a certain extent, it can represent the content of organic matter in sewage and play a crucial role in the growth and reproduction of microorganisms. At the same time, the chemical oxygen demand can also reflect the degree of water pollution, which is an important water quality index.

2.  $NH_4^+$;

The content of ammonia nitrogen is an important indicator in measuring water pollution, and excessive content in water discharges can lead to eutrophication. One of the functions of the aerobic tank is to nitrate ammonia nitrogen and oxidize it to nitrate nitrogen.

3. $NO_3^-$;

In the aerobic tank, nitrate nitrogen is produced by the nitration reaction of ammonia nitrogen. At this stage, ammonia nitrogen content in the water decreases while nitrate nitrogen content increases.

4. pH;

The growth and reproduction of microorganisms require a suitable pH range, and changes in pH can affect the absorption of nutrients and enzyme activity during microbial metabolism. In the actual sewage treatment process, the pH value is generally set within the range of 6–9.

5. T;

The activity of microorganisms is influenced by temperature, and high or low temperatures are not conducive to their growth. Generally speaking, when the temperature is around 25 °C, the growth of microorganisms is most favorable.

6. MLSS;

MLSS is an indicator of the amount of activated sludge in the reaction aeration tank and can be used as an indicator of the microbial biomass of activated sludge.

7. DO;

Dissolved Oxygen is the molecular form of oxygen present in water, which is a very important indicator in aerobic tanks and participates in nitrification reactions and organic matter degradation reactions. The concentration of dissolved oxygen can directly affect the effectiveness of sewage treatment.

8. HRT.

HRT is the average reaction time between sewage and microorganisms in the bioreactor, which largely determines the degree of sewage treatment.

In this paper, experiments were conducted using laboratory $A^2O$ wastewater treatment simulation equipment. After 180 days of experimentation, a total of 516 sets of experimental data were obtained, and after selecting, 500 valid data samples were retained. Specific data samples can be found in Table S1. Part of the sample data (collected in March 2023 in Northeast China) is shown in Table 3.

**Table 3.** Part of the content of the sampling sample.

| $I_{COD}$ | $I_{NH4^+}$ | $I_{NO3^-}$ | pH | T | MLSS | DO | HRT | $E_{COD}$ | $E_{NH4^+}$ | $E_{NO3^-}$ |
|---|---|---|---|---|---|---|---|---|---|---|
| 46.50 | 0.76 | 13.49 | 6.97 | 17.50 | 3.50 | 7.53 | 7 | 40.3 | 0.28 | 13.75 |
| 45.60 | 1.82 | 10.31 | 6.74 | 17.60 | 4.30 | 7.78 | 8 | 38.6 | 1.01 | 10.64 |
| 46.4 | 2.62 | 13.40 | 6.81 | 18.30 | 3.30 | 8.15 | 6 | 41.4 | 0.79 | 18.44 |
| 42.1 | 2.39 | 24.31 | 6.54 | 19.00 | 3.20 | 7.62 | 8 | 36.8 | 1.43 | 25.37 |
| 56.2 | 4.19 | 32.70 | 6.30 | 19.80 | 3.80 | 7.52 | 9 | 46.8 | 2.71 | 33.69 |
| 60.3 | 7.75 | 14.44 | 7.14 | 20.00 | 4.20 | 7.29 | 11 | 59.2 | 4.67 | 14.44 |
| | | | ⋯ | | | | | | ⋯ | |
| 40.5 | 4.35 | 4.70 | 7.08 | 20.60 | 4.52 | 7.02 | 13 | 39.2 | 1.66 | 5.43 |
| 41 | 7.65 | 13.66 | 6.93 | 19.50 | 4.21 | 7.14 | 12 | 38.7 | 5.27 | 12.58 |

### 2.2. Data Dimensionality Reduction Based on KPCA

Due to the large number of variables collected, in the process of modeling, the huge amount of computation will greatly reduce the efficiency of the algorithm and increase the calculation time. Therefore, it is necessary to reduce the dimensionality of the data and retain the maximum features of the sample as much as possible after dimensionality reduction. The data dimension reduction technology can improve the speed and efficiency of the algorithm while ensuring the accuracy of the algorithm.

Traditional dimensionality reduction techniques mainly use the Principal Component Analysis (PCA) method, but the function mapping of PCA in performing dimensionality reduction operations is linear, and many dimensionality reduction operations require nonlinear mapping in real tasks. In order to solve this problem, the kernel function was added before PCA dimensionality reduction, and the original data were mapped from low-dimensional space to high-dimensional space through the kernel function, so as to realize the feature division of high-dimensional space of data. Then, principal component analysis and dimensionality reduction were carried out to obtain the final feature information and realize the nonlinear dimensionality reduction of data, namely, the Kernel Principal Component Analysis method [25]. The common kernel functions are usually Gaussian, polynomial, exponential, and Laplace functions. In this paper, the Gaussian function is selected as the kernel function for dimensionality reduction. The KPCA dimensionality reduction process is as follows. The flow of dimensionality reduction using Matlab (R2021a) is shown in Figure 2.

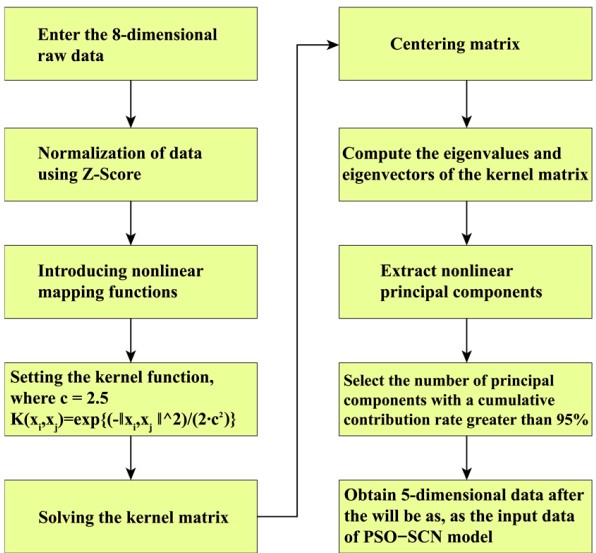

**Figure 2.** The Matlab implementation process of KPCA algorithm.

1.  Given a set of input vectors $X$ as the feature matrix, it is mapped to a high-dimensional space using a Gaussian kernel function, $R^d \rightarrow R^k$, $d < k$.

$$X = \begin{bmatrix} x_{11} & \cdots & x_{1n} \\ \ldots & \ddots & \ldots \\ x_{m1} & \cdots & x_{mn} \end{bmatrix} \in R^d \rightarrow \varphi(X) = \varphi\left(\begin{bmatrix} x_{11} & \cdots & x_{1n} \\ \ldots & \ddots & \ldots \\ x_{m1} & \cdots & x_{mn} \end{bmatrix}\right) \in R^k, \quad (1)$$

Among them, $m$ represents the number of feature value sample points, and $n$ represents the dimension of the measurement variable. $R^d$ is the $d$-dimensional space where vector $X$ is located, and $R^k$ is the $k$-dimensional space after high-dimensional mapping of vector $X$, $\varphi$ is the mapping function.

There exists a kernel function K such that: $K(x_i, x_j) = K(x_i{}^T x_j) = \varphi(x_i)^T \varphi(x_j)$, and K is a Gaussian kernel function.

$$K(x_i, x_j) = \exp\left\{ \frac{-\|x_i, x_j\|^2}{2 \cdot c^2} \right\}, \quad (2)$$

Among them, $x_i$ and $x_j$ are the sample data to be mapped, c is the Gaussian kernel function parameter, where the value of c is set to 2.5.

2.　Calculate the covariance matrix $C$ in the high-order space $R^k$.

$$C = \frac{1}{n}\sum_{i=1}^{n} \varphi(x_i)\varphi(x_i)^T, \tag{3}$$

3.　Calculate eigenvalues $\lambda$ and eigenvectors $\omega$ through iterative algorithms.

$$\varphi(X)\varphi(X)^T \omega = \lambda\omega, \tag{4}$$

4.　Obtain the projection of $x_i$ from high-dimensional space to low-dimensional space.

$$\omega = \sum_{i=1}^{n} \varphi(x_i)\alpha_i = \varphi(X)\alpha, \tag{5}$$

Combining the relationship between kernel functions K and $\varphi$, it is derived that $K\alpha = \lambda\alpha$, where $\alpha$ is the projection during feature space transformation, that is, the dimensionality reduced data.

5.　Sort the feature vectors according to the size of their eigenvalues, and take the dimensionality-reduced matrix P composed of the first few rows with a cumulative contribution rate greater than 95%.

Using the above steps to write the KPCA dimensionality reduction program in Matlab software, the relationship between the number of principal components and the contribution rate (Figure 3) was obtained by KPCA dimensionality reduction of the 500 sets of model input sample data in Table 3, and the main parameters of the KPCA program are shown in Table 4. The figure shows that when the cumulative contribution rate is greater than 95%, the data can be downgraded to five dimensions. Therefore, selecting the five-dimensional data obtained after dimensionality reduction as the input to the PSO-SCN model can reduce the model calculation time and improve the calculation speed.

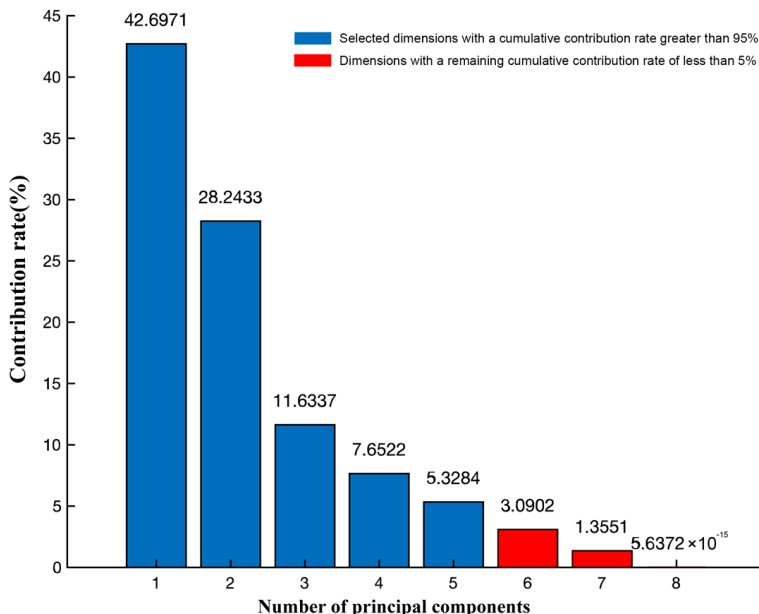

**Figure 3.** Contribution rate chart of each principal component.

**Table 4.** KPCA program main parameters.

| Symbol | Description | Value | Unit |
| --- | --- | --- | --- |
| c | Kernel function parameters | 2.5 | - |
| Con | Accumulated contribution rate | 95 | % |

### 3. Modeling of A$^2$O Aerobic Zone Based on PSO-SCN

*3.1. PSO Principle*

Particle Swarm Optimization Algorithm (PSO) is an evolutionary computational technique formed based on the behavior of birds feeding in flocks. The principle of the Particle Swarm Optimization algorithm is to find the optimal solution through the collaboration and information sharing between individual particles within the swarm. Particle swarm optimization algorithm has the advantages of small computational cost and fast convergence, so it is very convenient when used for models with strong real-time requirements [26,27].

The optimization process of the PSO optimization algorithm is as follows.

1.  Initialize the particle swarm. In D-dimensional space, each particle swarm has two attributes: velocity vector $V_i$ and position vector $X_i$. Random initialization is performed for $V_i$ and $X_i$.
2.  Obtaining the optimal position. The objective function is computed by the $V_i$ and $X_i$ of the particle to obtain the particle optimal position $P_{best}$ and the global optimal position $G_{best}$ in the space.
3.  Speed and position update.

Speed update Equation (6):

$$V_{i+1} = \omega * V_i + C_1 * rand_1 * \left(P_{best_i} - X_i\right) + C_2 * rand_2 * \left(G_{best_i} - X_i\right), \tag{6}$$

Position update Equation (7):

$$X_{i+1} = X_i + V_{i+1}, \tag{7}$$

where,$w$ is the inertia factor, representing the strength of the optimization ability; $rand_1$ and $rand_2$ are random numbers between [0, 1]; $P_{best}$, and $G_{best}$, represent the optimal position of the particles before updating and the global optimal position.

In the speed update, the dynamic $w$ can have better optimization performance than the static $w$. In this paper, the dynamic adjustment of adopts the linear decreasing weights (Equation (8)) strategy.

$$w_t = w_{max} - \frac{w_{max} - w_{min}}{T_{max}} t, \tag{8}$$

where $w_t$ is the inertia factor after $t$ iterations; $w_{max}$ and $w_{min}$ are the upper and lower limits of the inertia factor, generally taken as 0.9, 0.4; $T_{max}$ is the maximum number of iterations.

4.  After reaching the maximum number of iterations, obtain the final $P_{best}$ and $G_{best}$.

*3.2. SCN Principle*

Stochastic Configuration Network, originally proposed by Wang [28,29] et al. in 2017, is an incremental randomized neural network model. Unlike other randomized neural networks, in SCN, a supervised mechanism is used for parameter configuration of hidden nodes, which starts from a simple network with random input weights and biases, and uses the least squares method for the output weights and biases, which is able to construct a generalized forcer automatically and quickly. At the same time, SCN gives an inequality constraint to determine the error and thus reassign the input weights and biases. As the number of SCN nodes increases, the network is able to approximate any mathematical function and model. The SCN network is constructed with less manual intervention, has a strong generalized approximation performance, and achieves good results in solving classification and regression problems. The structure of SCN is shown in Figure 4. Suppose a set of samples $(X_i, Y_i)$ is given, $i = 1, 2, \cdots N$ ($N$ is a positive integer). Where $X = (x_1, x_2, \cdots x_N), x_i = \{x_{i,1}, x_{i,2}, \cdots x_{i,a}\} \in R^a$ is the input to the model and $Y = (y_1, y_2, \cdots y_N), y_i = \{y_{i,1}, y_{i,2}, \cdots y_{i,a}\} \in R^b$ is the output of the model. Assuming

that there are L nodes in the implicit layer in the SCN, construct the objective function $f_L : R^a \rightarrow R^b$.

$$f_L(X) = \sum_{j=1}^{L} \beta_j g_j \left( w_j^T, X, c_j \right), \tag{9}$$

where $w_j$ and $c_j$ are the weights and biases of the implicit layer, respectively; $g_j(\cdot)$ is the activation function of the $j_{th}$ node; $\beta_j = \left\{ \beta_{j,1}, \beta_{j,2}, \cdots \beta_{j,b} \right\}$ is the weight between the $j$th node of the implicit layer and the node of the output layer; and $f_L$ is the output function of the SCN network. The structural diagram of the SCN can be represented as follows. Among them, $X_i$ is the model input and $Y_i$ is the model output. The input layer has 1 to $A$ nodes, and the output layer has 1 to $B$ nodes. The SCN model is constructed by adding hidden layer nodes $L$.

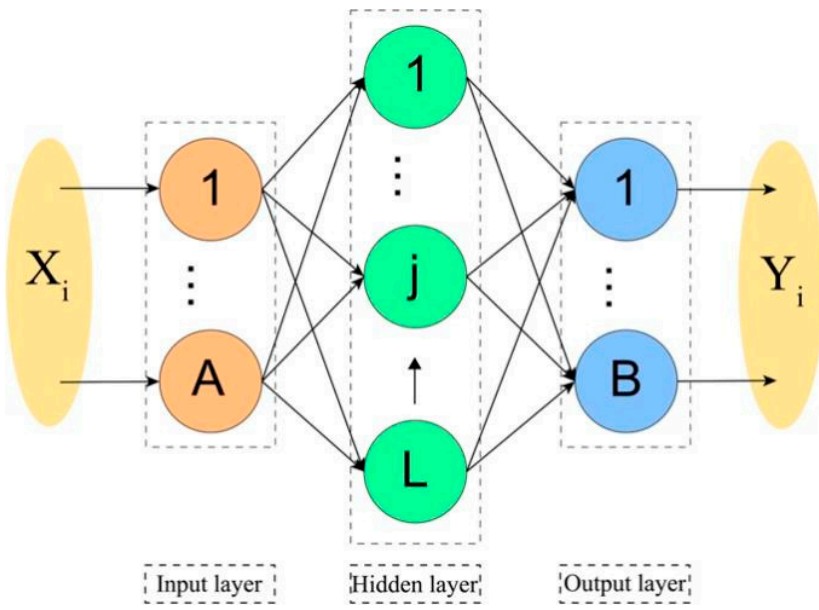

**Figure 4.** SCN Structure diagram.

*3.3. PSO Optimized SCN Model*

In this paper, the problem that SCN is not sufficiently optimized in the node parameters, which makes the model not accurate enough, is fully taken into account, and for the characteristics of the PSO algorithm, such as small computational cost and fast convergence, the PSO-SCN model is proposed, and PSO is used to optimize the weights and biases of SCN.

The optimization process is as follows.

1. According to the SCN characteristics, randomly initialize the weights and bias, and obtain the weight and bias matrices.
2. Take the error (Root Mean Square Error, RMSE) of the SCN model as the objective function, and use the PSO algorithm to optimize the weights and bias matrix, so that the error reaches the required range.
3. The optimized weights and bias matrices are put through the selection operation by the supervision mechanism to check whether the network output meets the requirements. If no, increase the hidden layer network nodes and return to step 2 to optimize the optimized weights and bias matrix again. If yes, end.
4. Output model.
5. The specific optimization process of the PSO-SCN model is shown in Figure 5.

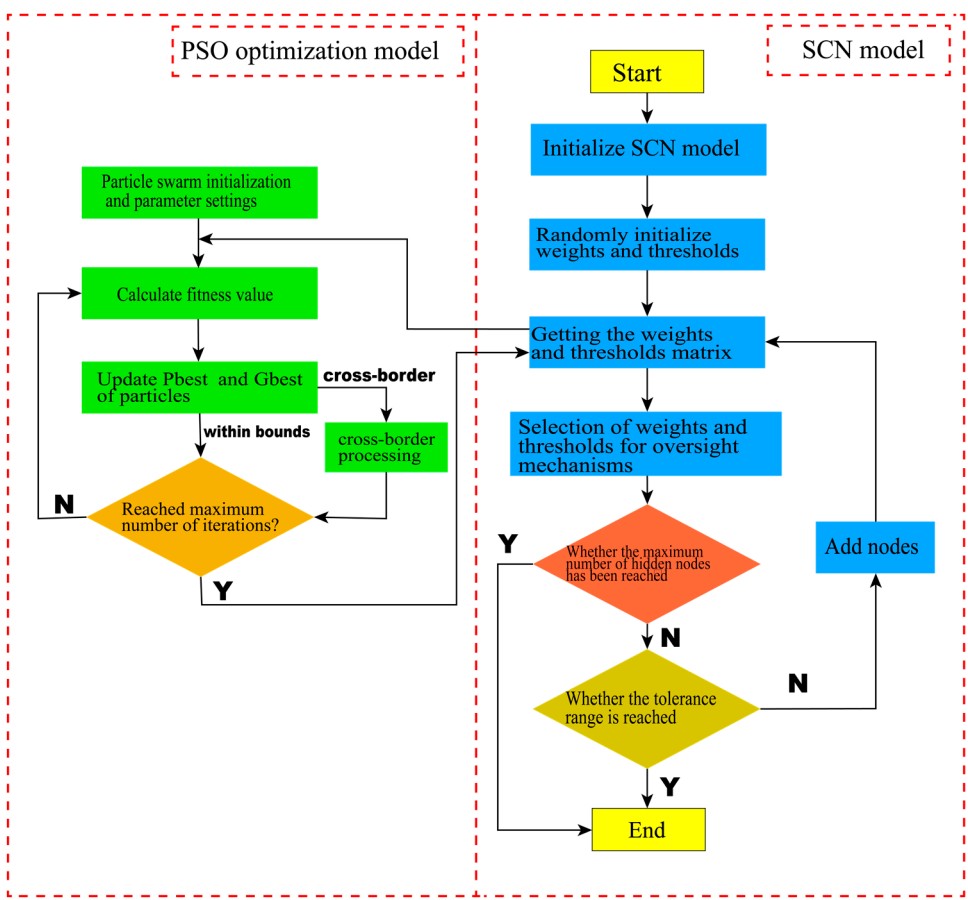

**Figure 5.** PSO-SCN algorithm flow chart.

## 4. Experimental Results and Analysis

In order to verify the effectiveness of the KPCA-PSO-SCN model proposed in this paper, after KPCA dimensionality reduction of 500 sets of experimental sample data, 80% of the samples are randomly selected as the training set and 20% as the test set to train and test the PSO-SCN model. According to the requirements of the article, the *RMSE* and *NSE* are used as the model accuracy measure. The *RMSE* formula (Equation (10)) is shown below.

$$RMSE = \sqrt{\frac{1}{N}\sum_{i=1}^{N}(y_{obs} - y_{mod})^2}, \ i = 1, 2, \cdots, N, \tag{10}$$

The *NSE* formula (Equation (11)) is shown below.

$$NSE = 1 - \frac{\sum_{i=1}^{N}(y_{mod} - y_{obs})^2}{\sum_{i=1}^{N}(y_{obs} - \overline{y_{obs}})^2}, \ i = 1, 2, \cdots, N, \tag{11}$$

Among them, $N$ is the number of samples; $y_{obs}$ is the true value, and $y_{mod}$ is the model output value.

Before training the PSO-SCN model, the training and test samples need to be normalized to eliminate the magnitude differences between the input and output data. The normalization process can process the data to the range of [0, 1], and the specific formula is shown in Equation (12).

$$x' = \frac{x - x_{min}}{x_{max} - x_{min}}, \tag{12}$$

Among them, $x$ is the original data; $x_{min}$ is the minimum value in the original data; $x_{max}$ is the maximum value in the original data; and $x'$ represents the normalized data.

The other key parameters of PSO-SCN are shown in Table 5.

**Table 5.** Key parameters of PSO-SCN model.

| | Model Parameter | Value |
|---|---|---|
| SCN | Maximum number of hidden nodes | 500 |
| | Maximum number of random configurations | 250 |
| | Training tolerance | 0.01 |
| | Random weight range | [0.5, 1,5, 10, 30, 50, 100, 150, 200, 250] |
| PSO | Particle swarm dimension | 10 |
| | Number of particles | 20 |
| | Evolutionary frequency | 50 |

Through the PSO-SCN model, the prediction of COD, $NH_4^+$, and $NO_3^-$ in the effluent of the aerobic zone was carried out, and the variation curves of the root mean square error of the training process of effluent COD, effluent $NH_4^+$, and effluent $NO_3^-$ concentrations in the aerobic zone, and the results of the model training and prediction were obtained, as shown in Figures 6–8. From Figures 6, 7 and 8a, it can be seen that the training error of the network gradually decreases with the increase in the structure, and when the number of nodes in the implicit layer is 285, 376, and 279, respectively, the root-mean-square error between the training set and the test set reaches less than 0.01, which satisfies the error tolerance set in the model. From Figures 6, 7 and 8b,c, it can be seen that the training error as well as the prediction error of the model for COD, $NH_4^+$, and $NO_3^-$ of the effluent water are small, and the fitting degree is high.

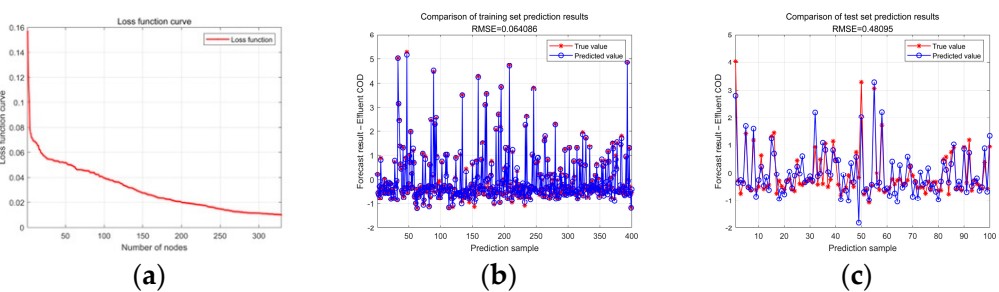

(a)      (b)      (c)

**Figure 6.** $E_{COD}$ concentration prediction. (**a**) Error plot; (**b**) training set RMSE; (**c**) testing set RMSE.

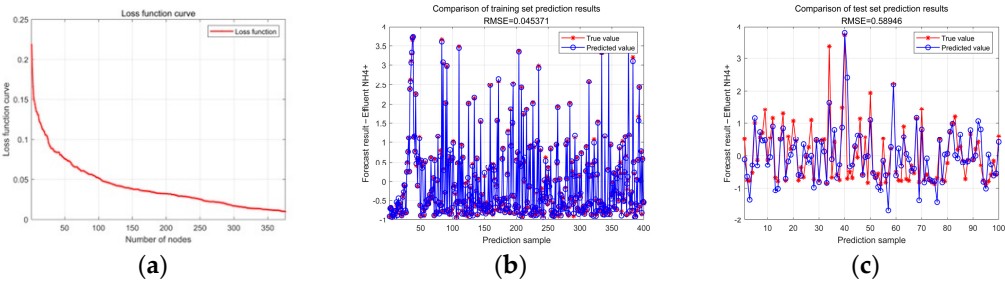

(a)      (b)      (c)

**Figure 7.** $E_{NH4^+}$ concentration prediction. (**a**) Error plot; (**b**) training set RMSE; (**c**) testing set RMSE.

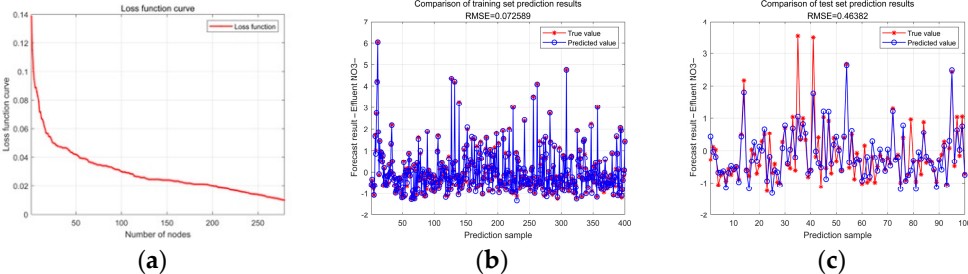

(a)      (b)      (c)

**Figure 8.** $E_{NO3^-}$ concentration prediction. (**a**) Error plot; (**b**) training set RMSE; (**c**) testing set RMSE.

In order to ensure the rigor of the algorithm and further evaluate the model performance, the classical water quality prediction algorithms such as SCN, PSO-BP, RBF, PSO-RBF, and PSO-SCN were selected for comparison with PSO-SCN. The specific results are shown in Table 6, which shows the comparison between the various types of algorithm models and the RMSE, NSE of each water discharge index, as well as the comparison information between the prediction time. Due to the fluctuation of program running results, the values in the table are selected from the results of 10 runs and the average is taken as the final result. From the results, it can be seen that the PSO-SCN model has a significant advantage over other algorithms in terms of prediction accuracy, except that the prediction accuracy of PSO-BP for $E_{COD}$ is better than PSO-SCN. However, comparing the prediction time, PSO-BP takes longer, and for models with stronger real-time requirements such as water quality prediction, models with smaller time are more suitable. Although RBF takes significantly less time than the PSO-SCN model, its prediction accuracy is larger than the PSO-SCN model. From the NSE results, the average predicted results for $E_{COD}$, $E_{NH_4^+}$, and $E_{NO_3^-}$ for each model are $-0.84991$, $0.760064$, $0.582304$, $0.51689$, and $0.606145$, respectively. From the average, it can be seen that PSO-SCN is superior to SCN, RBF, PSO-RBF, and lower than PSO-BP. However, combining RMSE and prediction time, the PSO-SCN model is more superior.

**Table 6.** Comparison of RMSE, NSE, and prediction time between different models.

| Model | RMSE | | | NSE | | | Prediction Time (s) |
|---|---|---|---|---|---|---|---|
| | $E_{COD}$ | $E_{NH_4^+}$ | $E_{NO_3^-}$ | $E_{COD}$ | $E_{NH_4^+}$ | $E_{NO_3^-}$ | |
| SCN | 2.3458 | 1.0974 | 1.6535 | $-1.51918$ | $-0.72963$ | $-0.30092$ | 8.83 |
| PSO-BP | 0.35598 | 0.61494 | 0.52357 | 0.85265 | 0.64262 | 0.784922 | 12.71 |
| RBF | 0.59504 | 0.82213 | 0.51919 | 0.846274 | 0.29246 | 0.60818 | 2.55 |
| PSO-RBF | 0.5821 | 0.7833 | 1.0526 | 0.60607 | 0.2546 | 0.69002 | 38.08 |
| PSO-SCN | 0.43999 | 0.58946 | 0.46382 | 0.72875 | 0.95011 | 0.72875 | 3.53 |

## 5. Conclusions

In this paper, in the study of the A$^2$O aerobic zone modeling method, a modeling method of the aerobic zone of the A$^2$O wastewater treatment process based on KPCA-PSO-SCN is proposed to realize the prediction of effluent COD, effluent $NH_4^+$, and effluent $NO_3^-$ concentration in the aerobic zone. The modeling method is characterized as follows:

1. Eight auxiliary variables that have a large impact on the effluent results were identified through the mechanistic reaction process in the aerobic zone. The KPCA method based on the Gaussian kernel function was used to downscale the sample data of the auxiliary variables, and the number of principal components with a cumulative contribution rate of more than 95% was selected to downscale the eight-dimensional data to five dimensions, which preserved the sample characteristics and improved the running speed and efficiency of the PSO-SCN algorithm.

2. For the characteristics of the SCN algorithm with less manual intervention and more general approximation, but insufficient optimization of node weights and biases, the particle swarm optimization algorithm with smaller computational cost and faster convergence speed is selected to optimize the weights and bias of the SCN algorithm. The effectiveness of the algorithm was verified by experimental data.

3. The classical water quality prediction algorithms such as SCN, PSO-BP, RBF, and PSO-RBF were compared, and the superiority of the PSO-SCN algorithm was verified in terms of RMSE, NSE and prediction time.

In this study, due to practical constraints, this process modeling only predicts the effluent indexes in the A$^2$O aerobic zone, and in future research, the whole process of wastewater treatment is considered to be modeled in order to achieve the clarity of the various stages and indexes in the A$^2$O process, and at the same time, it is also considered to combine the part of the A$^2$O wastewater treatment process that consumes the largest

amount of energy, i.e., the aeration energy, with the effluent water quality, and to create a multi-objective optimization model, which is used to guarantee the effluent quality at the same time and to ensure the quality of the effluent. It is used to minimize the aeration energy consumption while ensuring the effluent water quality.

**Supplementary Materials:** The following supporting information can be downloaded at: https://www.mdpi.com/article/10.3390/w15203692/s1, Table S1: A$^2$O aerobic zone experimental data.

**Author Contributions:** Conceptualization, X.T. and W.L.; methodology, X.T., Y.M. and Y.G.; software, W.L.; validation, W.L., L.L. and L.S.; resources, W.L.; data curation, W.L.; writing—original draft preparation, W.L.; writing—review and editing, W.L.; supervision, X.T.; project administration, X.T.; funding acquisition, X.T. All authors have read and agreed to the published version of the manuscript.

**Funding:** This research was funded by the Department of Science and Technology of Liaoning Province: 2021JH1/10400031.

**Data Availability Statement:** The datasets presented in this study can be obtained upon request to the corresponding author.

**Conflicts of Interest:** The authors declare no conflict of interest. The funders had no role in the design of the study; in the collection, analyses, or interpretation of data; in the writing of the manuscript; or in the decision to publish the results.

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
