# Peer review of "Modeling Method for Aerobic Zone of A2O Based on KPCA-PSO-SCN"

_water, doi:10.3390/w15203692_

Round 1

Reviewer 1 Report

The authors have tried to predict the WQI based on three significant water quality parameters: COD, NH4+, and NO3- with the help of bio-inspired optimization techniques like PSO. The paper at its present stage can not be accepted. However, if the article is improved by considering the following points then I can reconsider my decision :

1)Using only two factors(Error and Prediction time) to decide the performance of computer models is not accepted. Please include at least correlation and Nash Sutcliffe Efficiency to the already determined Error Metrics.

2)Why only Aerobic Zone? This needs to be explained in detail.

3)Authors have utilized PSO. This type of optimization technique often falls into local minima. How to avoid or recover from this was not mentioned.

4)What is the novelty of this paper? What are the Novel findings of this paper? 

5)Stochastic Configuration Network Why? RBF Why and PSO Why? As there are many better algorithms available in the World. So justification of the using of such algorithms is required.

6)SCN works with supervisory data. No description of the training and testing data was provided. So how the SCN model was trained? How the model learned the problem.

Try to answer the above points I will reconsider my decision.

Author Response

Dear Reviewer 1:

We greatly appreciate your professional comments on our manuscript (2620170). As you are concerned, we do have several issues that need to be addressed. Based on your suggestion, we have made extensive corrections to the previous manuscript. Your comment is listed in Italics below, and our response is provided in normal font. The revised parts of the manuscript have been highlighted in yellow.

1、Using only two factors(Error and Prediction time) to decide the performance of computer models is not accepted. Please include at least correlation and Nash Sutcliffe Efficiency to the already determined Error Metrics.

Thank you very much for your professional advice. Based on your suggestion, we have added NSE to determine the predictive performance of the model based on the RMSE and prediction time. NSE is usually applied in hydrological models to evaluate the quality of model simulation results. However, in neural network models, various indicators such as RMSE,MAE are usually used as evaluation criteria to determine the quality of the model, among which RMSE is more commonly used. After adding NSE according to your suggestion, the completeness of the article is reflected.

Revision location: The NSE formula has been added on line 319 of the manuscript (Page 11). The NSE prediction results have been added in lines 352-365 and Table 6 of the manuscript (Page 12). And modifications were also made in the summary section, line 386 (Page 13).

2、Why only Aerobic Zone? This needs to be explained in detail.

I will explain to you from the following three aspects.

  1. a) Modeling the A2O aerobic zone is part of my research on overall modeling of the A2O process and optimization of aeration energy consumption. Our subsequent work includes modeling the Anaerobic zone, Anoxic zone, overall process modeling (prediction of effluent quality), and optimization of aeration energy consumption in the A2O process. The modeling of A2O aerobic zone is the top priority of the entire research.
  2. b) The Aerobic zone is the most complex and variable part of the biochemical reactions and influencing variables in the entire A2O process, and the biochemical effect of the Aerobic zone has a crucial impact on the effluent quality. Our further research includes the prediction of effluent quality, so it is of great significance to focus on describing and modeling the Aerobic zone.
  3. c) The aeration energy consumption in the Aerobic zone accounts for approximately 60% of the total energy consumption of the sewage treatment plant. Modeling the Aerobic zone also lays the foundation for subsequent research on aeration energy consumption optimization, and has significant practical significance.

3、Authors have utilized PSO. This type of optimization technique often falls into local minima. How to avoid or recover from this was not mentioned.

In the 3.1 PSO Principle section of the manuscript, PSO has defined and used inertia weight values. The gradual reduction of linear inertia weights (Formula 8, Page 8-9, lines 261-263) can effectively solve the problem of PSO being prone to local optima and improve its optimization performance.

4、What is the novelty of this paper? What are the Novel findings of this paper? 

In the manuscript, the main novel was the combination of PSO and SCN to create the PSO-SCN model, which was first applied to the modeling of sewage treatment processes. From Table 6 (Page 12), it can be seen that the performance of PSO-SCN is superior to traditional RBF, PSO-BP, PSO-BRF models.

5、Stochastic Configuration Network Why? RBF Why and PSO Why? As there are many better algorithms available in the World. So justification of the using of such algorithms is required.

SCN, as a relatively new neural network technology, has the unique advantages of strong universal approximation performance and less manual intervention compared to other neural networks. And currently, this technology is rarely applied in modeling sewage treatment processes. Previous researchers often used BP and RBF as basic techniques in modeling wastewater treatment processes, and optimized and improved them. Therefore, we modeled the sewage treatment process based on the advantages of SCN and the characteristics of low computational cost and fast convergence of PSO. We compared it with RBF, PSO-BP, and PSO-RBF, and found that PSO-SCN outperformed other technologies in prediction accuracy, prediction time, and NSE.

In future work, we will consider using more optimization techniques to optimize SCN, explore the best optimization techniques, and compare them with more algorithms.

6、SCN works with supervisory data. No description of the training and testing data was provided. So how the SCN model was trained? How the model learned the problem.

We have provided a total of 50 pieces of 10% of all data in the form of supplementary materials for your reference (Table S1. xlsx). Each data contains 8 input variables and 3 output variables. Part of the data can be seen in Table 3. 80% or 400 randomly selected as training data and 20% or 100 selected as test data.

After KPCA, the training process can be seen in Figure 5 (Page 10):

  1. a) Firstly, randomly initialize and obtain the weights and biasmatrices in the SCN model.
  2. b) Next, PSO is used to optimize the weights and biasfor the first time and assign them to the SCN weight and bias
  3. c) In this case, input 400 pieces of data into the SCN model and compare the output results with the actual data to verify whether RMSE is within the set range.
  4. d) If the set conditions are met, it ends and the SCN model is obtained. If the set conditions are not met, repeat steps a, b, and c until the set conditions are met.
  5. e) Subsequently, input the test data into the model, check the RMSE between the output results and the actual data under this weight and bias, and output.

Thank you again for your suggestion. We have made every effort to modify it. If you have any other questions, please feel free to let me know.We will provide a very detailed explanation. Wishing you a pleasant life.

Yours Sincerely,

Wenxia Lu and other authors

Reviewer 2 Report

Current work used modelling of A2O process for its optimization as the most widely used sewage treatment process in urban areas is the Anaerobic-Anoxic-Oxic (A2O) sewage treatment process. Therefore, modeling the sewage treatment process and predicting the effluent quality are of great significance. A process modeling method based on Kernel Principal Component Analysis - Particle Swarm Optimization - Stochastic Configuration Network(KPCA-PSO-SCN) is proposed for A2O aerobic wastewater treatment process. Firstly, eight auxiliary variables were determined through mechanism analysis, including Chemical Oxygen Demand(COD) and ammonia nitrogen(NH4+) and nitrate nitrogen(NO3-) of influent water, pH, temperature(T), Mixed Liquor Suspended Solid(MLSS), Dissolved Oxygen(DO) and hydraulic residence time(HRT) in the aerobic zone. Dimensionality reduction was carried out using the kernel principal modelling.

Current work needs correct spacing mistakes. Figures captions and legend need to be larger fonted. P values need to be added to justify results .

No essential novelty at different conditions can be drawn. As clearly seen also by Fig.

Intro part should be 1.5 pages.

The data that support the findings of this study should added with the ref as per format.

Figs have strange trends with high fluctuations,maybe parallel tests and error bars needed.

Biology reactions as a promising and viable option for new treatment have been successfully done and characterized, and could be included: 

https://doi.org/10.3390/w13212959 https://doi.org/10.3390/w14132063  https://doi.org/10.1080/09593330.2012.665487

Ok

Author Response

Dear Reviewer 2:

We sincerely appreciate your valuable feedback on our manuscript (2620170), which is very useful for improving the quality of the article. Based on your suggestion, we have made every effort to revise the manuscript. Your comment is listed in Italics below, and our response is provided in normal font. The revised parts of the manuscript have been highlighted in green.

  • Current work needs correct spacing mistakes. Figures captions and legend need to be larger fonted.P values need to be added to justify results .

We are so sorry for our negligence of those mistakes. We have adjusted the spacing for the format issue you raised. According to the template I am using, the font for the Figures captions and legend is Palatino Linotype 9. I am not sure if there is a problem with the template I am using. I have requested the correct template from the editor.

We believe that your suggestion to add a P-value to determine the research results is very good. However, due to laboratory operating conditions, we will consider adding the value of P in future work. This article focuses on the prediction of COD, NH4+, and NO3- in the effluent of aerobic zones.

  • No essential novelty at different conditions can be drawn. As clearly seen also by Fig.

In the manuscript, the main novel was the combination of PSO and SCN to create the PSO-SCN model, which was first applied to the modeling of sewage treatment processes. From Table 6 (Page 12), it can be seen that the performance of PSO-SCN is superior to traditional RBF, PSO-BP, and PSO-BRF.

The trend of the results in the graph has been given in your fifth opinion, and the reason for the fluctuation is due to normalization. From the perspective of fitting degree, the model has a good fit, and the RMSE between the simulated output and the actual data is very small.

  • Intro part should be 1.5 pages.

We have added content to make the introduction section 1.5 pages (Page 2). The specific content added is highlighted in green.

  • The data that support the findings of this study should added with the ref as per format.

Our data comes from the laboratory, totaling 500 pieces. We currently provide you with 50 pieces of data as a reference. Each data contains 8 input variables (MLSS, COD, NH4+, NO3-, T, PH, DO, HRT) and 3 output variables (COD, NH4+, NO3-). We will provide the data in the form of supplementary materials (Table S1. xlsx). And cited in line 182 (Page 5) of the manuscript.

  • Figs have strange trends with high fluctuations,maybe parallel tests and error bars needed.

In Figures 6-8 (b) and (c) (Page 12), the horizontal axis represents the predicted sample and the vertical axis represents the result of normalization.

The specific reason for the seemingly significant fluctuations is that after KPCA dimensionality reduction, the data was normalized to within the range of [0,1]. The subsequent data training and testing operations resulted in the output results being within a small numerical range. Making data that was not originally highly volatile appear to be highly volatile.

But the ultimate goal is to obtain the degree of model fitting, so it is not meaningful to perform reverse normalization on the data after KPCA dimensionality reduction and normalization.

  • Biology reactions as a promising and viable option for new treatment have been successfully done and characterized, and could be included: https://doi.org/10.3390/w13212959 https://doi.org/10.3390/w14132063  https://doi.org/10.1080/09593330.2012.665487

We have cited the articles you provided (Page 1). The Biology reactions technology used in these three articles is very advanced and has great reference value for our article. We will also consider using the Biology reactions technology mentioned in the above for in-depth exploration of our wastewater treatment research in future work.

Thank you again. If you have any further questions, please feel free to contact us and we will provide the most detailed explanation. Wishing you a pleasant life.

Yours Sincerely,

Wenxia Lu and other authors

Round 2

Reviewer 1 Report

I have no further comments as the authors have answered to all my queries.

May be improved

Reviewer 2 Report

Thanks for revision. MS has improved. Check spacings after and before references.

Provide Intro middle section also with references from newer than 5 years.

Check everywhere ref style to be according to journal, mistaken:

"Kusiak, Caneta, Bagheri et al. [20-22] "

Font sizes in figures need to be larger as figures itself ie 6-9 .

Add spacings and st deviations to tables, three lined preferred.

Start capital letters in conclusions and elsewhere.

good